# Involvement of Actin in Autophagy and Autophagy-Dependent Multidrug Resistance in Cancer

**DOI:** 10.3390/cancers11081209

**Published:** 2019-08-20

**Authors:** Magdalena Izdebska, Wioletta Zielińska, Marta Hałas-Wiśniewska, Alina Grzanka

**Affiliations:** Department of Histology and Embryology, Faculty of Medicine, Nicolaus Copernicus University in Toruń, Collegium Medicum in Bydgoszcz, 85-092 Bydgoszcz, Poland

**Keywords:** autophagy, actin, cancer, multidrug resistance

## Abstract

Currently, autophagy in the context of cancer progression arouses a lot of controversy. It is connected with the possibility of switching the nature of this process from cytotoxic to cytoprotective and vice versa depending on the treatment. At the same time, autophagy of cytoprotective character may be one of the factors determining multidrug resistance, as intensification of the process is observed in patients with poorer prognosis. The exact mechanism of this relationship is not yet fully understood; however, it is suggested that one of the elements of the puzzle may be a cytoskeleton. In the latest literature reports, more and more attention is paid to the involvement of actin in the autophagy. The role of this protein is linked to the formation of autophagosomes, which are necessary element of the process. However, based on the proven effectiveness of manipulation of the actin pool, it seems to be an attractive alternative in breaking autophagy-dependent multidrug resistance in cancer.

## 1. Introduction

In the case of normal cells, autophagy is a mechanism used for the degradation of unnecessary or abnormal proteins, dysfunctional organelles and intracellular pathogens. This prevents the accumulation of unwanted particles in the cell and contributes to the internal homeostasis. What’s more, the process allows the release of macromolecules necessary for the synthesis of new proteins [1].

Many studies indicate a link between autophagy impairments and many diseases such as neurodegenerative diseases, lung diseases or sepsis [2,3,4]. Numerous reports also confirm the relationship between autophagy and cancer [5,6,7,8]. However, in this case, the role of the process is not strictly defined. Due to current literature reports, it is suggested that in the initial stages of carcinogenesis autophagy is one of the ways of body’s defense as the pathway of abnormal cells death [9]. This does not allow the further development of the disease and is favorable. At the same time, in the case of higher-grade cancers, there is a close link between tumor development and the occurrence of autophagy [10]. This is due to the fact that for rapidly dividing cancer cells, the key aspect is an access to the right amount of nutrients. If they cannot be delivered through the blood supply, decomposition of whole cells or they components through autophagy can be a source of nourishment for neighboring cells. Furthermore, autophagy can be a defense mechanism against environmental stress. This is probably one of the pathway of tumor cell resistance to chemotherapy [11]. This is confirmed by studies conducted on the non-small cell lung cancer A549 line, in which the use of an autophagy inhibitor overcame the resistance of cells to 5-fluorouracil (5-FU) but also in the case of colorectal cancer in the in vitro and in vivo models [12,13]. For these reasons, targeting the autophagy process is a promising alternative in the treatment of cancer. As it has been proven, actin is one of the elements necessary for this process to occur, for example through its involvement in the formation and maturation of autophagy vesicles [14,15,16]. Therefore, the manipulation of the actin pool can be a susceptible way to overcome the multidrug resistance of many types of cancer. In this review we focus on the participation of actin in the autophagy process and its correlation with autophagy-dependent multidrug resistance in cancer. In addition, we describe potential therapeutic targets associated with the inhibition of cytoprotective autophagy through actin manipulation.

## 2. Autophagy

Autophagy (the term was used from 1860s but described by Christian de Duve in 1963) at first was a process observed in yeast, however, it turned out that many stages and factors can also be applied to mammalian cells [17,18]. In 2016, Japanese scientist Yoshinori Ohsumi received the Nobel Prize for discovering the molecular mechanisms underlying autophagy [19]. In the context of cancer progression, the authors proved the inverse relationship between autophagy and malignant potential as the deletion of one of the proteins involved in autophagy–Beclin-1 led to the enhanced spontaneous, as well as, hepatitis B virus–induced mutations rate [20].

Much earlier, in the 90s, autophagy-related proteins (ATGs) were described as a factor playing the key role in transport of cargo to the autophagic vacuole in yeast [21]. These studies were of great importance in understanding the role of the process, not only in many physiological, but also pathological states. However, ATGs are not only a key element of autophagy, but also perform other important functions in the cell, associated with inter alia (i.a). cytokine secretion (ATG7) or phagocytosis (ATG9, ATG16L1). Furthermore, the role of ATGs in cell proliferation (ATG3, -5, -7) has also been demonstrated, which may further influence the participation of these proteins in the development of diseases such as cancer [22,23,24]. More studies involving genetic mutations disrupting specific autophagy/non-autophagy functions of ATGs are necessary to elucidate their role [25,26].

There are three types of autophagy: Chaperone-mediated autophagy (CMA), microautophagy and macroautophagy. In CMA, proteins in the complex with chaperones (such as heat shock cognate 70—Hsc-70) are translocated across the lysosome membrane after prior binding to lysosome-associated membrane protein type 2A (LAMP-2A) receptor [27]. Microautophagy is a direct uptake of cytoplasmic elements by lysosomes. In turn, in macroautophagy, the cytoplasmic cargo is delivered to the lysosome by autophagosom (a double membrane-bound vesicle) which bind together to form autophagolysosomes [28]. Due to the large interest in macroautophagy and its role in cancer, this review focuses only on this mechanism and in the further parts of the article the word autophagy is used exactly in this context. 

In a physiological sense, the autophagy process is crucial in establishing cell homeostasis. It acts as an intracellular quality control system, which prevents accumulation of unnecessary or damaged cellular proteins and organelles [29]. In addition, this process is activated in response to stimuli such as: Lack of nutrients, pathogens or oxidation, allowing the cell to adapt to new conditions [30]. Moreover, mutations of genes encoding proteins involved in the regulation of autophagy induce carcinogenesis processes. In addition, the dual nature of autophagy is a very interesting issue, because paradoxically in the case of cancer, on the one hand it may lead to tumor suppression, but on the other it promotes cancer [31]. Regardless of the nature of autophagy, it takes place in several stages: Initiation and phagophore nucleation, elongation and autophagosome formation, fusion with lysosome, and autophagolysosome formation with degradation of intravesicular elements [32,33]. The first step is connected with unc-51-like kinase 1 (ULK1), also known as ATG1, complex activation. Although five ULK homologues are known (ULK1, ULK2, ULK3, ULK4 and STK36) just ULK1 and ULK2 are involved in autophagy signaling and only ULK1 in essential in this process [34]. This involved in the formation of autophagosome complex consists of serine/threonine protein kinase-ULK1, ATG13 and ATG101 and focal adhesion kinase family interacting protein of 200 kDa-FIP200 (also known as retinoblastoma coiled coil protein 1-RB1CC1) (Figure 1) [35,36,37,38]. Dissociated from mechanistic target of rapamycin complex 1 (mTORC1) ULK1 complex undergoes autophosphorylation and induces the phagophore nucleation at phagophore-formation site (PAS) triggering activation of the nucleation factor class III phosphatidylinositol-3-kinase (PI3KC3) complex I [39,40]. This multipart structure comprises PI3K vacuolar protein sorting 34 (VPS34), ATG14 and Beclin-1, which is phosphorylated by ULK1 and acts as a scaffold for the PI3K complex. Additionally, PI3KC3 complex I also contains activating molecule in BECN1-regulated autophagy protein 1 (AMBRA1), which binds Beclin-1, stabilizes the complex and mediates linking to the cytoskeleton and p115-general vesicular transport factor. Elements of the described complex affect phosphorylation of phosphatidylinositol and generate phosphatidylinositol-3-phosphate (PI3P) the production of which occurs at endoplasmic reticulum structure/ region called omegasome (Figure 1) [41,42]. Subsequently PI3P recruits two effectors: WD repeat domain phosphoinositide-interacting protein 2 (WIPI2) and double FYVE containing protein 1 (DFCP1) in order to reshape the omegasome and allow the further isolation of phagophore membrane [43,44]. This step involves the presence of the third complex which consists of ATG16L1, ubiquitin-like ATG12 and ATG5 [45]. WIPI2 is directly bound to ATG16L1 causing the recruitment of the other complex elements and thereby to ATG3, which catalyzes the conjugation of the ubiquitin-like ATG8 family to the phosphatidylethanolamine (PE) in the membrane of growing phagophore. This group of proteins contains i.a., microtubule-associated protein 1 light chain (LC3) and γ-aminobutyric acid receptor-associated proteins (GABARAP) [44,46]. Additionally, ATG4 cleaves pro-LC3 to form LC3-I, while ATG7 and ATG3 participate in conversion of LC3-I into its membrane-anchored form LC3-II [47,48]. The ATG8 family members play important role in autophagy, not only because they participate in the elongation and closure of phagophore membrane but are also involved in cargo recognition in LC3-interacting region (LIR) (Figure 1) [49,50,51,52]. The final steps of the autophagy include autophagosome maturation and fusion with lysosome. Created structure (autophagolysosome) contains elements which undergo degradation (Figure 1). 

Characteristic proteins involved in the process usually become its markers. In the case of autophagy, these are Beclin-1 and LC3-II. Due to the complexity of autophagy, there are many potential targets to modulate the process [53,54]. Of course, the most popular proteins are described above complexes, but the elements of cytoskeleton may also be an excellent target for the manipulation (positive or negative). One of them is actin, which involvement in autophagy is still not well known.

## 3. Actin Cytoskeleton and Its Engagement in Autophagy Process

The cytoskeleton builds one of the most complex cell structures necessary for nearly all processes occurring in its area [55]. One of the main cytoskeletal proteins is actin. Although it is commonly associated with muscle cells, it also occurs in non-muscle ones where it can be from 1 to even 5% of all cell proteins [55]. On the basis of the isoelectric point three main actin isoforms are distinguished (alpha [α], beta [β] and gamma [γ]). The isoforms present in muscles are α-skeletal, α-aortic smooth, α-cardiac, γ2-enteric smooth while in non-muscle cells β- and γ1-cytoplasmic forms occurs [56]. The protein participates in many cellular processes mainly related to muscle contraction and cell mobility, but it also takes part in movements of organelles and vesicles, cell division, cytokinesis, cell signaling, maintenance of cell junctions and cell shape, cell cycle and cell death [55,56,57,58,59,60,61,62,63]. In muscle cells actin with myosin form the basic units of muscle contraction called sarcomere. Polymerized form of actin is a main component of filopodia and lamellipodia, which allows migration. Cortical actin is involved in cell shape maintenance. Additionally, this protein is necessary during cytokinesis as contractile ring and apoptosis in the apoptotic blebs formation. Different forms of actin are not only observed in cytoplasm but also in the cell nucleus, where they interact with RNA polymerases and chromatin remodeling complexes [55,64]. Furthermore, literature reports indicate its key function in autophagy e.g., by engagement in the formation of autophagosomes [65]. The crucial role of actin in cell functioning is highlighted by its conservativeness. It can be found in all eukaryotic representatives (except for nematode sperm). Moreover, in the prokaryotic organisms an analogue of the protein is present in the form of MreB protein [66].

Actin in a cell may occur in two forms of globular G-actin or fibrillar F-actin in both the nucleus and cytoplasm [67]. The monomeric form is evenly distributed over the entire cell volume, whereas the accumulation of the polymeric form is closely related to the state of the cell [68]. The actin polymerization process starts with the creation of a stable nucleation center that is normally formed by three G-actin monomers in a spontaneous event driven by formin. The next step is the elongation of the filament by adding successive G-actin units to the both ends of the nucleation center, which is a reversible process. As the concentration of G-actin units locally drops, one end of the filament grows (barbed-end) while the other decreases in length (pointed-end) leading to a phenomenon called treadmilling. Elongation continues until the rate of the loss of ADP-actin from the pointed-end is greater than ATP-actin assembly, when the filament begins to shrink [68]. The dynamics between two actin forms and locations determines the functioning of the cell. At the same time, any changes in the actin structure require the involvement of actin-binding proteins (ABPs). ABPs can both stabilize the structure of filaments, e.g., through capping or anchoring in cell membranes, as well as prevent polymerization by binding to monomers [67]. Additionally, filaments can be cut into shorter fibers or create more complex structures by introducing cross-links or forming bunches. One such protein is Arp2/3, which serves as nucleation site for new actin filaments that bind to the already existing filament at the angle of 70 degrees. The Arp2/3 complex consists of seven subunits such as ARPC1, ARPC2, ARPC3, ARPC4, ARPC5, ARP2 and ARP3, which are connected with actin nucleation, filament polymerization and branch formation [69]. The complex is also involved in autophagy process through its participation in autophagosome formation [65].

Furthermore, actin plays a significant role in death at the cellular level [55]. It is known that the described protein participates among others in autophagy (Figure 2) [15]. The involvement of actin in the formation and maturation of autophagy vesicles has been known since the 90’s, but the exact mechanism of its participation was unclear [15,16]. At the beginning of the studies on the engagement of actin in autophagy, it was observed that its reorganization (mainly polymerization) affects the first stages of the described process [16]. The next step was characterization of the colocalization of microfilaments with main autophagy markers such as LC3, ATG14 and Beclin-1 [14]. What is more, F-actin was localized also on autophagosomes isolated membranes together with DFCP1, ATG5, ATG16 and ULK1(Figure 2) [70]. Factors that indirectly engage actin in the autophagy process are ABPs like Arp2/3.

Many reports present the role of nucleation promoting factors (NPFs) in the activation of the Arp2/3 complex, which in turn is connected with actin cytoskeleton. Different NPFs, despite diverse structures, all contain C-terminal proline-rich and Wiskott-Aldrich homology 2 (WH2) domain, which mediates their interactions with profilin-actin, actin, and the Arp2/3 complex [40]. In mammalian autophagy three NPFs are defined. First of them is localized mainly in the cytoplasm junction-mediating and regulatory protein (JMY), which is regarded as actin nucleation promoting factor. There are three domains in its structure. Domain on the C-terminal region allows binding and activation of the Arp2/3 complex [71]. In turn, N-terminal region contains LIR which allows the interactions with the autophagy marker LC3. It is also known, that mutation of LIR causes lack of JMY in autophagosome, reduces levels of LC3-II and precludes the formation of actin structure. This allows to conclude that JMY participates in the accumulation of actin in autophagic membrane (Figure 2) [71]. Hu et al. took a step further and reported that JMY-dependent microfilaments polymerization on phagophore and autophagosome membranes is regulated by stress-responsive activators p300 (STRAP) and LC3 [72]. The membrane-bound LC3 drafts JMY and stimulates actin network formation [73].

The second NPF is WASP homolog-associated protein with actin, membranes and microtubules (WHAMM), which like JMY is responsible for Arp2/3-dependent actin nucleation (Figure 2) [74]. Kast and co-workers in 2015 presented the research which suggests that the WHAMM-dependent actin-comet tail mechanism is directly connected with the Arp2/3 complex activation [75]. This conclusion was confirmed by the inhibition of actin tail formation after the down-regulation of WHAMM expression. The same results were obtained after blocking of Arp2/3 complex or its interaction with described NPF. The scientists showed also the very important role of this NPF in autophagosome biogenesis and its colocalization with the autophagy markers LC3, DFCP1 and p62 [74,75,76]. Additionally, size and number of autophagosomes depend on WHAMM expression and localization of WHAMM and actin on the border of neighboring autophagosomes [75].

The last, poorly studied, but equally important for the proper course of autophagy is Wiskott-Aldrich syndrome protein and SCAR homologue (WASH), which activates the Arp2/3 and induces actin network formation in vesicle during endocytosis [77,78,79]. On the other hand, it is also known, that this NPF can be the negative regulator of autophagy through inhibition of VPS34 activity, which causes blocking of Beclin-1 ubiquitination [79,80].

Additional factors regulating the activity and functioning of actin during autophagy are small GTPases of the Rho superfamily. These proteins modulate F-actin formation via GTPase signaling cascades. Small GTPases are connected with Arp2/3-dependent actin nucleation and upon starvation the actin regulatory protein RhoA plays part in autophagosome formation [16]. Additionally, Ser/Thr Rho kinase 1 (ROCK 1) binds and phosphorylates Beclin-1 and in this way promotes autophagy [81]. Small GTPases also include the Rac family proteins among which three isoforms are distinguished-Rac1, Rac2 and Rac3. Zhu and co-workers presented Rac3 as a negative regulator, because its knockdown allows induction of autophagy [82]. Similar results were presented by Aguilera’s team, who observed the negative effect of Rac1 in starvation-mediated autophagy [16]. One of the essential elements in the autophagosomes formation is localized at the plasma membrane small GTPase-ADP-ribosylation factor 6 (Arf6). This protein participates in the generation of phosphatidylinositol 4,5-bisphosphate (PIP2) and colocalizes with ATG16L1 [83].

Different factor, which regulates the participation of actin in the autophagy is ATG9. It is a transmembrane protein trafficked from Golgi apparatus to phagophore [84,85]. Many studies indicate that ATG9 participate only in the first step of autophagy, as it is probably removed from the autophagosome before its closure, what explains its absence on the autophagosomal and lysosomal membrane [85]. In mammalian cells, upon starvation, the translocation of this protein is very important for autophagosomal biogenesis and requires participation of ABPs. These proteins are i.a., Annexin A2, spire-type actin nucleation factor 1 (Spire1) or mentioned earlier Arp2/3 and WASH complex [86]. After down-regulation of Annexin A2 and Arp2/3 the accumulation of actin around ATG9A vesicles is decreased, but in the case of the regulation of Spire’s actin nucleation we still do not know much [86,87]. The results suggest that microfilaments may play a pivotal role in omegasome maintenance and next in formation of phagophore [70]. The shape of the double-membrane autophagosomal vesicle is also regulated by actin nucleation.

On the other hand, inhibition of actin filaments growth by actin-capping proteins is also observed. Actin on the autophagosomal membrane collocates not only with the marker of autophagy-LC3, but also with the proteins connected with regulation of actin dynamic such as cofilin, CapZ (actin capping protein) or Arp2/3. CapZ was identified as the negative regulator of actin polymerization, because it blocks the attachment of G-actin monomers to the growing filament [65]. In the lamellipodia, where the formation of a long microfilaments is required, CapZ is disassociated by phosphoinositide—PI(4,5)P_2_ from growing end of actin filaments [88]. The same mechanism was presented in autophagy, however, here omegasomes are structures rich in the PI(3)P [70]. Additionally, the pool of PI(3)P depends on VPS34 activation and stimulates other factors, which influences autophagosome maturation [89]. It is undeniable that the actin cytoskeleton and its reorganization are necessary for proper formation of autophagosome. However, many reports indicate also that actin is required during fusion of the autophagosome with the lysosome. First step of this process is migration of autophagosomes to the lysosomes location [90,91]. Lee and co-workers presented histone deacetylase-6 (HDAC6) as a factor which induces cortactin-dependent, actin-remodeling machinery in nutrient-independent autophagy. Polymerization of G-actin enables fusion of autophagosome with lysosome and protein aggregates digestion. Remarkably, use of the actin polymerization inhibitor (Latrunculin A) reduced the fusion of described elements [91]. In turn, Hong et al. confirmed participation of phosphatidylinositol 3,5-bisphosphate (PI(3,5)P_2_) in actin dynamic regulation [92]. Based on the mentioned studies, Hasegava et al. described inositol polyphosphate-5-phosphatase E (INPP5E) as a novel autophagy regulator [93]. They presented the new mechanism and cascade reaction of the last steps of autophagy. The authors suggest that at the beginning, INPP5E catalyzes PI(3,5)P_2_ conversion to PI(3)P, on the lysosomes what activates cortactin. In turn, cortactin binds with polymerized actin and consequently leads to its stabilization, which is necessary during fusion of autophagosome with lysosome (Figure 2) [93]. Furthermore, if PI(3)P is connected with WASH complex, then INPP5E indirectly regulates the fusion process by affecting actin polymerization [93].

Although, the exact role of autophagy process in cancer is still not fully elucidated in general it supports the progression of established tumors. This is caused by reduction in their sensitivity to intrinsic and extrinsic stimulation which under standard conditions would promote cell death [94]. This is confirmed by a growing evidence indicating that autophagy inhibition often limits the proliferation and metastatic potential of malignant cells. Furthermore, high grade human tumors in general are characterized by increased autophagic flux, which correlates with an invasive/metastatic phenotype and poorer prognosis [95].

## 4. Autophagy in Multidrug Resistance

Drug resistance remains one of the most serious problems of oncology. It is estimated that approximately 90% of chemotherapy failures are associated with tumor invasion and metastasis related to this phenomenon [96]. Tumor resistance to cytostatics can be either inherent or acquired as a result of the mutation created during the administration of chemotherapy. As the mechanism of drug resistance (DR) in cancer, several factors may be distinguished like reduction in drug uptake, enhanced drug efflux by membrane transporters (mostly ATP-binding cassette (ABC) transporters) or changes in its metabolism [97]. In turn, multidrug resistance (MDR) is defined as the resistance of tumor cells to several cytostatics with the different structures and mechanisms of action. Inhibition of apoptosis e.g., by mutation in the p53 gene, adaptive changes through epigenetic regulation, mutations of drug therapeutic aims or alterations in the tumor microenvironment may also contribute to the development of MDR. As MDR is responsible for most of the chemotherapy failures, its inhibition would be of great importance for oncological patients [97]. Recent literature reports indicate the process of autophagy may, in at least some cases, be the basis of MDR, but the relationship between these phenomena is still not fully understood [98,99,100,101,102,103,104]. Like in the initiation and progression of cancer also in the case of MDR, the role of autophagy may be twofold. 

Many literature reports indicate the involvement of autophagy in drug resistance. As shown by Shuhua et al. in the case of colorectal cancer, tumor tissues were characterized by significantly higher expression of autophagy-related genes such as Beclin-1, LC3, and Rictor, which levels positively correlated with the level of multidrug resistance protein (MDR-1) gene expression [105]. This indicates the participation of autophagy in the development of MDR. Prosurvival role of autophagy was also confirmed in breast, ovarian, esophageal, lung, prostate, renal and pancreatic cancers [98,99,100,101,102,103,104]. Similar effect was observed in the case of glioma and bladder cancer [106,107]. In many of these studies, the use of an autophagy inhibitor was associated with the increase in sensitivity to the cytostatics used. According to research carried out by Pan et al. and Du et al. autophagy blocking in colorectal and non-small cell lung cancers cells treated with 5-fluorouracil (5-FU) using a 3-methyladenine (3-MA) autophagy inhibitor, enhances cell apoptosis through increased production of reactive oxygen species [12,108]. The efficiency of this method has also been confirmed for the chloroquine (CQ), an autophagy inhibitor which efficacy and safety of use have been approved by the Food and Drug Administration. The result showed that additional administration of CQ enhanced limitation of tumor cell growth caused by 5-FU in both in vitro and in vivo conditions [109,110]. However, literature reports indicate the benefits of this approach also in the use of other cytostatics such as paclitaxel, doxorubicin, cisplatin or epirubicin [97]. Moreover, as was proven by Yang et al. MDR may be a direct consequence of autophagy process [106]. As the scientists showed, both the doxorubicin and vincristine treatment of human leukemia cells led to an increase in the level of S100A8, which was necessary for the formation of the Beclin-1–class III phosphatidylinositol 3-kinase (BECN1-PI3KC3) complex, and thus contributed to enhanced autophagy rate in the result of which the drug resistance developed. This conclusion was confirmed by modulation of the S100A8 level, as the resistance raised with increased S100A8 expression, while the reduced MDR was observed after lowering the transcript level [106]. Moreover, commonly occurring in the solid tumors hypoxia was distinguished as a factor leading to the intensification of autophagy and thus also drug resistance. In the case of human bladder cancer cells both silencing of hypoxia-inducible factor-1 (HIF-1) and use of 3-MA resulted in enhanced apoptosis rate [107].

However, the matter is not as simple as it may seem. At the same time, many studies indicate the involvement of autophagy in the response of tumor cells to cytostatics as a mechanism of cell death [111]. One of the interesting studies is the work published in 2014 in the International Journal of Oncology by Kaewpiboon et al. in which it was proved that a natural compound is able to overcome drug resistance by inducing autophagy which in turn stimulates the process of apoptosis [112]. Furanocoumarin representative Feroniellin A in etoposide-resistant A549 cells (A549RT-eto) led to both a downregulation of P-glycoproteins but also a significant increase in LC3 II, Beclin-1 or ATG5. The observed changes were also associated with the enhanced apoptosis. Moreover, the inhibition of autophagy by using siRNA for Beclin-1 caused a reduction in apoptosis, while the addition of the autophagy inducer was associated with an additional increase in the percentage of apoptotic cells [112]. Researchers assessing the effect of caffeine on PC12D cells (cellosaurus cell line) have reached similar conclusions. Also, in this case, the treatment of the cells with the compound led to both the intensification of the autophagy process, but also apoptosis. In turn, the inhibition of autophagy resulted in the reduction of apoptosis [113]. However, the induction of autophagy without an increase in apoptosis may also be responsible for the inhibition of tumor cell proliferation. This was confirmed by studies with the use of metformin and myeloma cells (RPMI8226 and U266 lines) in which the enhanced expression of autophagy markers was observed in cells treated with the compound, and at the same time no apoptosis intensification was noted [114]. The fact that even the same type of compound in various cases can cause extremely different effects is an additional confusion. The previously mentioned metformin, which induced autophagy as a mechanism of cell death in myeloma cells, in breast cancer cells has a cytoprotective character [115]. However, the results may change with the addition of other factors. As was shown by Yeo et al. combination of autophagy inhibitor spautin-1 together with metformin may enhance the compound action in the model of BRCA1-deficient mice [116]. It should also be noted that not only cytostatisc are autophagy inductors, but other factors applied must be carefully considered. Our previous research shown, that the use of lidocaine in rat glioma C6 cell line leads to cytoprotective autophagy [117].

## 5. Autophagy as a Therapeutic Target in Cancer

Although the complexity and multistageness of the autophagy process makes it difficult to fully understand the underlying mechanism, it also creates many options for its regulation in a both positive and negative manner. So, the question remains whether to inhibit or promote autophagy? The answer is probably it depends. One of the factors that determines the role of autophagy in the process of cancer development may be its stage [5]. However, it is certainly not the only determinant. Probably this issue must be considered individually depending on the tumor type and the cytostatic used. For several reasons, this review will focus rather on the inhibition of autophagy. First of all, although many of the studies indicate the induction of autophagy by various cytotoxic drugs or compounds of natural origin in most cases, it is a rather cytoprotective autophagy, and its inhibition results in an increase in the sensitivity of tumor cells to a factor applied [99,100,101,102,103,104,105,106,107]. Secondly, although researches on autophagy inductors are underway, most of them are working nonspecifically, also impairing other cellular processes such as mTOR signaling [5]. Finally, and thirdly, this work focuses mainly on the involvement of actin in the autophagy process and their connection with cancer. Although it is a fairly recent topic, several works indicate that also actin and ABPs may be a therapeutic target, but rather in inhibition than promotion of autophagy. However, we are also aware of the evidence that it is the induction of autophagy that can yield the best results by the promotion of the further response of immune system [118]. To overcome these problems, a tumor vaccine containing the inhibitor of autophagy and proteasomes was designed (NCT03057340) [119]. Highly expressed by tumor tissues defective ribosomal products (DRiPs) and short-lived proteins (SLiPs) are known for their properties of inducing anti-cancer activity of the immune system. However, under standard conditions these proteins are rapidly degraded by proteosomes. Inhibition of proteasomes results in an increase in the concentration of the indicated proteins. DRiPs and SLiPs accumulate in autophagosomes; however, the inhibition of autophagy does not allow their further degradation in autophagolysosomes. The increase in DRiPs and SLiPs concentrations stimulates the response of the immune system, [119]. The one thing we know for sure is that choosing the right approach is definitely not a simple matter and requires further analysis.

One of the types of factors most widely used to inhibit autophagy are chemical inhibitors. Among this group, many substances may be distinguished such as Bafilomycin A1, 3-MA, CQ and hydroxychloroquine (HCQ). Factors with such properties also include trehalose and caloric restriction mimetics or physical exercises [120]. Of these substances, only CQ and HCQ are recognized by the FDA as safe to use. The both compounds works through deacidifcation of lysosomes and block the fusion of autophagosomes with lysosomes. However, HCQ is mainly used for clinical trials due to its lower toxicity [120]. During numerous preclinical and clinical studies, it was proved that combining HCQ together with various cytostatics may give promising results in the case of many types of cancers, e.g., pancreatic, renal, colorectal, myeloma and melanoma [121,122,123,124,125]. However, it has been observed that in combination with alkylating drugs such as, for example, temozolomide may result in increased toxicity and Grade 3/4 neutropenia and thrombocytopenia [126]. Another problem is that although at low doses administered over a short period of time, there was no significant increase in the incidence of undesirable symptoms in comparison to the cytostatics alone, however, increasing the dose and prolonging treatment time may result in serious adverse effects. One of them was retinal toxicity [127]. After five years of combined therapy, it is recommended that patients undergo annual examines to detect early changes in the retina. In addition, pharmacodynamic analyzes have shown that doses up to 1200 mg/d may result in only a poor in vivo effect, while low pH significantly limits cellular uptake of the compound, making it difficult to use since acidic extracellular pH is one of the characteristics of tumors. It drives the search for new substances that inhibit autophagy [123]. Among others created so far are inhibitors directed against autophagy regulators such as VPS34, ULK1 and ATG4B. One of the promising substances is also Lys05, a homologue of HCQ that also causes deacidification of lysosomes. However, as demonstrated by the study, this compound is able to affect the tumor also administered in monotherapy and not only in combination with cytostatics [123]. The ideal inhibitor, however, should not only have a high efficiency but also be selective or susceptible to modifications that enable selectivity, which can reduce side effects. In studies carried out on the mouse model, it was shown that global inhibition of autophagy through the *Atg7* gene knockout resulted in the death of all animals due to neuronal toxicity [128]. Additionally, Karsli-Uzunbas et al. has showed that in healthy mice deletion of the *Atg7* gene leads to severe changes such as immunodeficiency and neurodegeneration, however, in mice suffering from non-small cell lung cancer acute inhibition of autophagy was associated with effective restriction of tumor growth and was not lethal. [129]. An alternative approach is to use manipulation of genes associated with autophagy. 

As demonstrated by Gong et al., the use of siRNA directed against ATG7 in combination with docetaxel in the case of MCF-7 breast cancer cell line. Combined therapy resulted in up to 2.5-fold increase in cytotoxicity and apoptosis rate. At the same time, the micellar system used by the researchers allowed for a significantly selective targeting of tumor tissues [130]. In turn, Zhang et al. proved that in the case of patients with gastric cancer in the tumor cells, increased expression of ATG5 and acid-sensing ion channels (ASICs) occurs, while the silencing of the gens resulted in the limiting of the proliferation of SGC-7901 cells. In addition, the use of shRNA against the ATG5 and ASICs genes was associated with an extended survival time and a reduction in tumor size in murine heterotopic xenograft model [131]. The use of siRNA against ATG5 and ATG7 also resulted in a decrease in autophagy and intensified the anti-cancer effect of cisplatin drug-resistant esophageal cancer cells [132]. The effect of autophagy inhibitors may also be affected by the stage in which the inhibition occurs. This is confirmed by studies conducted on prostate cancer cell line, in which the blocking of autophagosome turnover caused an enhanced necroptosis rate, while limiting the maturation of autophagosomes resulted in a decrease in necroptosis [133]. In this context, it becomes attractive to manipulate the actin pool, which participates in virtually every stage of the autophagy process, which results in a large flexibility of this approach. The actin pool can be regulated by affecting the ABPs involved in the rearrangements of proteins necessary for the occurrence of autophagy. One of such ABPs is Arp2/3 allowing the creation of branched actin networks, which plays a pivotal role in omegasome formation, shaping of autophagosome membrane and autopfagosome-lysososme fusion. As was proven by Moreau et al. both pharmacological inhibition with CK-666 (Arp2/3 inhibitor) and manipulation of Arp2 levels with siRNA resulted in a reduction in the number of autophagosomes observed [86]. Moreover, the study confirmed that Annexin A2 together with actin regulates ATG9A sorting from endosomes. The potential benefits of lowering the Arp2/3 protein level are also highlighted by the fact that the increase in its expression is observed in many types of cancer and linked with poorer prognosis for oncological patients [134].

Another therapeutic target for autophagy inhibition may be HDAC6. Although the protein does not directly participate in the activation of autophagy, it is probably responsible for the fusion of the lysosome with autophagosom due to the recruitment of the cortactin-dependent actin remodeling complex that causes the actin cytoskeleton to rearrange during the fusion which ends with autophagolysosome formation [91]. Research conducted by Lee et al. indicates that the actin used for polymerization comes mainly from the pool involved in the formation of autophagosome and not from the structure of the lysosome. Moreover, the HDAC6 knockout resulted in autophagy blocking at the stage of fusion of autophagosome with the lysosome, which in the case of mice led to accumulation of abnormal proteins in brain structures and, as a result, to neurodegeneration [91]. However, in the case of mice suffering from T-cell acute lymphoblastic leukemia, treatment with vincristine together with inhibition of HDAC6 led to an increase in the anti-cancer activity of the drug but was also associated with the alleviation of one of the cytostatic side effects-sensory neuropathy [135]. What’s more, Kaliszczak et al. demonstrated that both the use of a selective HDAC6 inhibitor C1A as well as the genetic knockout of the gene result in suppression of autophagy and to further inhibition of cell growth in the case of malignant cells (Myc-positive neuroblastoma cells, KRAS-positive colorectal cancer and multiple myeloma) [136].

The results presented the participation of actin cytoskeleton in the early steps of autophagy indicated that using the actin depolymerizing agents (Latrunculin B and Cytochalasin B) under starvation inhibits the MAP1LC3 formation, which numbers correlates with the autophagosomes. In turn the stabilization of microfilaments will not affect the autophagy process in any way [16]. The same authors also point to the role of actin dynamics regulators (Rho family proteins) in describing process. After knockdown of RHOA protein accumulation of MAP1LC3 is impeded too.

Although available studies proved that manipulation of the actin pool through ABPs effectively inhibits the autophagy process, according to our knowledge, it has not yet been investigated how this procedure will affect the survival of drug-resistant cancer cells treated with cytostatics. However, information included in our article suggest that it may be a promising method of autophagy inhibition that allows to conquer autophagy of cytoprotective character and thus, to overcome such phenomena as MDR.

On the other hand, some recent literature reports indicate that actin manipulation may be characterized by pleiotropic effects [137]. Although the idea of manipulating the actin structure is nothing new, its use is limited by the cytotoxic effect observed after administration of substances targeting this cytoskeletal element. This is due to the lack of distinction between actin in cancer cells and that in muscle cells, which leads to dysfunction of both cell type [138]. It is associated with, among others, such effects as cardiotoxicity. For this reason, compounds such as cytochalasin D and jasplakinolide, despite their proven antiproliferative activity against cancer cells, did not pass the preclinical tests. However, targeting actin regulating proteins in a manner specific for cancer cells may be the answer in this case. As demonstrated by studies carried out by Stehn et al. antitropomyosin compound TR100 preferentially targets non-muscular tropomiosin-containing filaments [138]. This resulted in a reduction in the proliferation rate of neural crest-derived tumor cells with no cytotoxic effect on myocardial cells in both in vitro and in vivo tests [138]. Similar conclusions were obtained from the research carried out by Foerster et al. using the actin hyperpolymerizating agent-Chondramide, which in the case of breast cancer was characterized by high tumor cell specificity [139]. Interestingly, the use of selective inhibitors of the Rho family of small GTPases representative-cell division cycle protein 42 (Cdc42) may also be an effective tool for breast cancer therapy [140]. Cdc42 serves as one of the key regulators of actin dynamics by coordinating signals promoting actin polymerization through effector proteins such as WASP or Transducer of Cdc42-dependent actin assembly (TOCA) families [140]. Moreover, WASP serves as a direct link between Cdc42 and Arp2/3, which, due to the important role in autophagy, potentially makes this protein an attractive therapeutic target also in this context. However, despite promising preclinical results, currently no Cdc42 targeting compound is used in clinical trials.

## 6. Conclusions

Although, the association of autophagy with the cancer initiation and progression is unquestionable its role in the process remains a mystery. That’s why choosing a treatment strategy is so difficult. In this context, methods with high flexibility appear to be the most promising. One of the key factors involved in the autophagy process at virtually all stages is actin. Due to its engagement in many steps of the process, manipulation of the actin pool, seems to be a very interesting approach due to its wide potential applications. The next step in this research field should be the assessment of the combined effect of actin pool manipulation, e.g., by regulating the expression of ABPs along with treatment with substances of cytostatic properties.

## Figures and Tables

**Figure 1 cancers-11-01209-f001:**
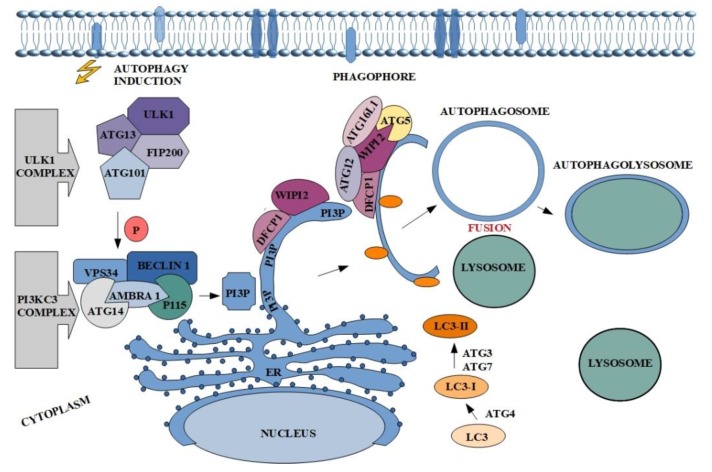
The course of the autophagy along with the main proteins involved in the process. The description of the figure can be found in the text above.

**Figure 2 cancers-11-01209-f002:**
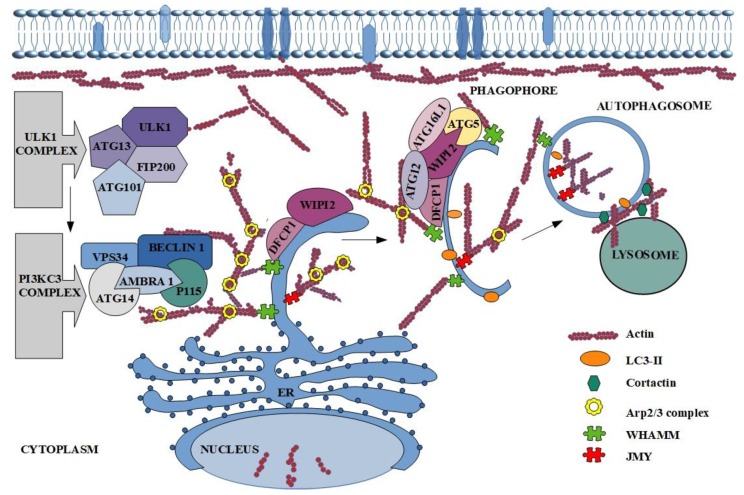
The participation of actin in autophagy process. The description of the figure can be found in the text above.

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
