# Peer review of "Involvement of Actin in Autophagy and Autophagy-Dependent Multidrug Resistance in Cancer"

_cancers, 2019, doi:10.3390/cancers11081209_

Round 1
Reviewer 1 Report
Major concerns
1. > 40% of references are reviews. Mainly in the first three chapters the majority of works cited are reviews. Statements or opinions about the described biological processes seem to be rather taken from these reviews than from original works.
2. The figures in the review are too small and too dark. Names of proteins cannot be read and the text only refers only sporadically to the figures, although they are highly needed for clarification of the described processes.
3. The first part of the review (part 1 – 3) is written in a very confusing style, is not well structured, and remains unclear to the reader. I recommend reworking of all three sections for clarification, style, and restriction to relevant parts only.
4. The connection between autophagy regulating processes and actin cytoskeleton regulation is not obvious after reading of the manuscript.
5. Organization of the actin cytoskeleton and its regulation is not well described. I would suggest a proper introduction to actin cytoskeleton function, organization and regulation, and clear definition of proteins involved in both autophagy and actin cytoskeleton regulation that will be mentioned later in the manuscript.
6. Chapter 3 has no comprehensive structure and describes involvement of NPF with autophagy in (confusing) detail without any conclusion.
7. Many references are missing.
8. Unclear sentence structure mainly in the first three chapters which does not seem to be connected to the following part of the review.
9. The review’s objective should be clarified.
Minor concerns:
1. Line 56: explanation of abbreviation Hsc-70 missing.
2. Line 144: mentioning of the role of the actin cytoskeleton in apoptosis seems out-of-context. Especially with regard to the plethora of cellular functions the actin cytoskeleton is involved with.
3. Line 188: Wiskott-Aldrich syndrome protein and SCAR homologue is WASH and not N-WASp.
4. Role of actin cytoskeleton in fusion of autophagosome with lysosome not detailed enough (line 238).
5. Several times the authors refer to the term “actin pool” that was never defined before or is known in the literature.
6. Studies to target the actin cytoskeleton have been conducted with the result of high cardiotoxicity. While the authors refer to past and ongoing clinical trials involving autophagy inhibitors, (recent) literature regarding targeting the actin cytoskeleton in cancer is completely missing (e.g. Zhang et al., 2019, Cells; Foerster et al., 2014, Cell Death Dis.; Stehn et al., 2013, Can Res)
Author Response
Dear Reviewer 1
Thank you for all your valuable comments. All of the issues were carefully considered and the appropriate changes were introduced. They include:
Major concerns
> 40% of references are reviews. Mainly in the first three chapters the majority of works cited are reviews. Statements or opinions about the described biological processes seem to be rather taken from these reviews than from original works.- Whenever possible, the literature cited was exchanged for original papers.
The figures in the review are too small and too dark. Names of proteins cannot be read and the text only refers only sporadically to the figures, although they are highly needed for clarification of the described processes.- The figures were improved and more refers to the figures in the text were added.
The first part of the review (part 1 – 3) is written in a very confusing style, is not well structured, and remains unclear to the reader. I recommend reworking of all three sections for clarification, style, and restriction to relevant parts only.- All three sections were extensively revised.
The connection between autophagy regulating processes and actin cytoskeleton regulation is not obvious after reading of the manuscript.- Manuscript was redrafted to clarify the connection between actin and autophagy.
Organization of the actin cytoskeleton and its regulation is not well described. I would suggest a proper introduction to actin cytoskeleton function, organization and regulation, and clear definition of proteins involved in both autophagy and actin cytoskeleton regulation that will be mentioned later in the manuscript.- Additional information were added for clarification purpose.
Chapter 3 has no comprehensive structure and describes involvement of NPF with autophagy in (confusing) detail without any conclusion.- NPFs were described because of their interactions with the Arp2/3 complex, and thus also indirectly with the actin cytoskeleton, as specified in the text. However, unnecessary details were erased.
Many references are missing.- The literature was improved and the missing references were added.
Unclear sentence structure mainly in the first three chapters which does not seem to be connected to the following part of the review.- In our opinion, the chapters on actin and autophagy are necessary in the work describing the connection of both of these elements. However, the text was extensively revised
for greater consistency.
- The review’s objective was added in the introduction.
Minor concerns:
Line 56: explanation of abbreviation Hsc-70 missing.- The explanation of abbreviation Hsc-70 was added.
Line 144: mentioning of the role of the actin cytoskeleton in apoptosis seems out-of-context. Especially with regard to the plethora of cellular functions the actin cytoskeleton is involved with.- Unnecessary elements were removed from the text.
Line 188: Wiskott-Aldrich syndrome protein and SCAR homologue is WASH and not N-WASp.- It was corrected.
Role of actin cytoskeleton in fusion of autophagosome with lysosome not detailed enough (line 238).- The role of actin cytoskeleton in fusion of autophagosome with lysosome was explained in more detail.
Several times the authors refer to the term “actin pool” that was never defined before or is known in the literature.- Term actin pool was used in the literature before (10.1007/BF02780546, 10.1038/ncb3641, 10.1242/jcs.113241, 10.1242/jcs.203760, 10.1091/mbc.E11-06-0582, 10.1091/mbc.e11-06-0582, 10.1083/jcb.201705216) so it was not exchange in the main text.
Studies to target the actin cytoskeleton have been conducted with the result of high cardiotoxicity. While the authors refer to past and ongoing clinical trials involving autophagy inhibitors, (recent) literature regarding targeting the actin cytoskeleton in cancer is completely missing (e.g. Zhang et al., 2019, Cells; Foerster et al., 2014, Cell Death Dis.; Stehn et al., 2013, Can Res). The paragraph on the suggested topic was added as recommended. Best regards, Magdalena IzdebskaReviewer 2 Report
Line 34 “In the case of higher-grade cancers, there is a close link between tumor development and theoccurrence of autophagy [10] “
More explanation is needed.
“As it has been proven, actin is one of the elements necessary for his process to occur”
Please explain how the actin is involved. The big picture. I am aware that one section on Actin is thereafter.
“Just three years ago, in 2016, Japanese scientist Yoshinori Ohsumi received the Nobel Prize for discovering the molecular mechanisms underlying autophagy [17]. “
What we have learned from his study regarding cancer progression in the follow up studies ?
General comment
“Autophagy molecular mechanism has been explained, a link to cancer progression should be emphasized (Line 47-107)”
Line 239-241 “Numerous examples of actin involvement in autophagy and cooperation with many regulators of this process, make this protein the perfect target of therapy during the protective autophagy of cancer cells.”
Sentence is diffused. What have we learnt about Cancer progression autophagy and actin involvement?
Author Response
Dear Reviewer 2
Thank you for all your valuable comments. All of the issues were carefully considered and the appropriate changes were introduced. They include:
Comments and Suggestions for Authors
Line 34 “In the case of higher-grade cancers, there is a close link between tumor development and the occurrence of autophagy [10] “
More explanation is needed.
The clarification was added.“As it has been proven, actin is one of the elements necessary for his process to occur”
Please explain how the actin is involved. The big picture. I am aware that one section on Actin is thereafter.
Appropriate information was added.“Just three years ago, in 2016, Japanese scientist Yoshinori Ohsumi received the Nobel Prize for discovering the molecular mechanisms underlying autophagy [17]. “
What we have learned from his study regarding cancer progression in the follow up studies?
General comment
General comment concerning cancer progression in the context of Yoshinori Ohsumi works was added.“Autophagy molecular mechanism has been explained, a link to cancer progression should be emphasized (Line 47-107)”
The link between autophagy and cancer progression was highlighted in additional part of the text.Line 239-241 “Numerous examples of actin involvement in autophagy and cooperation with many regulators of this process, make this protein the perfect target of therapy during the protective autophagy of cancer cells.”
Sentence is diffused. What have we learnt about Cancer progression autophagy and actin involvement?
The sentence was deleted.Best regards,
Magdalena Izdebska
Reviewer 3 Report
Izdebska et al. present a well-structured review on the potential of targeting actin to inhibit autophagy, largely to overcome multidrug resistance in cancer. The topics covered are comprehensive and informative for the reader. Several suggestions to improve the review are as follows:
1) The authors should emphasize that the non-autophagy functions of ATGs could be the reason for confounding roles of these genes in diseases such as cancer (PMID:31199916). Accordingly, more specific genetic mutants (knock-in) which disrupt specific autophagy/non-autophagy functions of ATG genes will be necessary to dissect these functions (PMID: 30403914, 29849149).
2) Lines 305-307, additional observations of autophagy inhibition acting cooperatively with metformin treatment have been observed (PMID:29938573).
3) Lines 364-366, although chronic inhibition of autophagy maybe detrimental, it has been shown that acute inhibition of autophagy may effectively restrain tumor growth and is not lethal (PMID: 24875857).
4) Inhibition of actin may have pleiotropic effects, which could potentially increase toxicities. It would be useful to balance this review by providing some discussion on this.
Author Response
Dear Reviewer 3
Thank you for all your valuable comments. All of the issues were carefully considered and the appropriate changes were introduced. They include:
The authors should emphasize that the non-autophagy functions of ATGs could be the reason for confounding roles of these genes in diseases such as cancer (PMID:31199916). Accordingly, more specific genetic mutants (knock-in) which disrupt specific autophagy/non-autophagy functions of ATG gene will be necessary to dissect these functions (PMID:30403914, 298491149).- As suggested, the fragment concerning ATG proteins was supplemented with the proposed information.
Lines 305-307, additional observations of autophagy inhibition acting cooperatively with metformin treatment have been observed (PMID:29938573)- As suggested, the fragment concerning metformin treatment was supplemented with the proposed information.
Lines 364-366, although chronic inhibition of autophagy maybe determental, it has been shown that acute inhibition of autophagy may effectively restrain tumor growth and is not lethal (PMID:24875857).- Mentioned information was added to the main text.
Inhibition of actin may have pleiotropic effects, which could potentially increase toxicities. It would be useful to balance this review by providing some discussion on this.- The paragraph on the suggested topic was added as recommended.
Best regards,
Magdalena Izdebska
Round 2
Reviewer 1 Report
The manuscript has been significantly improved and most of my concerns have been addressed, at least to some extent.
A few additional suggestions:
Line 135: the reference “55” only refers to actin in apoptosis, additional references should be provided for the other actin-driven processes. Line 236 “Actin polymerization takes places by adding… “ please delete this sentence. In general, the English remains relatively poor and I suggest that the authors get editing help from someone with full professional proficiencyin English.
Author Response
Dear Reviewer
Thank you for all your additional suggestions. All of the issues were carefully considered and the appropriate changes were introduced. They include:
Additional articles provided for the other actin-driven processes were added and the literature has been supplemented with appropriate items. The sentence „Actin polymerization takes places by adding… “ was deleted. Grammar, spelling and punctuation has been checked by a native English speaking colleague and all changes were marked in the text.
Best regards,
Magdalena Izdebska